# The Opposing Functions of Protein Kinases and Phosphatases in Chromosome Bipolar Attachment

**DOI:** 10.3390/ijms20246182

**Published:** 2019-12-07

**Authors:** Delaney Sherwin, Yanchang Wang

**Affiliations:** Department of Biomedical Sciences, College of Medicine, Florida State University, 1115 West Call Street, Tallahassee, FL 32306, USA; delaney.sherwin@med.fsu.edu

**Keywords:** protein phosphatase, chromosome bipolar attachment, kinetochore, spindle assembly checkpoint

## Abstract

Accurate chromosome segregation during cell division is essential to maintain genome integrity in all eukaryotic cells, and chromosome missegregation leads to aneuploidy and therefore represents a hallmark of many cancers. Accurate segregation requires sister kinetochores to attach to microtubules emanating from opposite spindle poles, known as bipolar attachment or biorientation. Recent studies have uncovered several mechanisms critical to chromosome bipolar attachment. First, a mechanism exists to ensure that the conformation of sister centromeres is biased toward bipolar attachment. Second, the phosphorylation of some kinetochore proteins destabilizes kinetochore attachment to facilitate error correction, but a protein phosphatase reverses this phosphorylation. Moreover, the activity of the spindle assembly checkpoint is regulated by kinases and phosphatases at the kinetochore, and this checkpoint prevents anaphase entry in response to faulty kinetochore attachment. The fine-tuned kinase/phosphatase balance at kinetochores is crucial for faithful chromosome segregation during both mitosis and meiosis. Here, we discuss the function and regulation of protein phosphatases in the establishment of chromosome bipolar attachment with a focus on the model organism budding yeast.

## 1. Introduction

In mitosis, chromosome segregation occurs in a way that each daughter cell receives one copy of every chromosome. This process is vital to all cell life, and several mechanisms ensure that sister chromosomes are attached by microtubules emanating from opposite spindle poles to establish bipolar attachment. The kinetochore is a complex of dozens of proteins arranged on the centromere of a chromosome, and it mediates chromosome attachment to microtubules [1]. In budding yeast, each kinetochore is attached to a single microtubule, unlike mammalian cells, in which each kinetochore is attached to multiple microtubules [2]. The establishment of chromosome bipolar attachment, or biorientation, enables the onset of anaphase. The spindle assembly checkpoint (SAC) is activated to delay anaphase entry until every chromosome has made a bipolar attachment. The failure of chromosome bipolar attachment and compromised SAC function results in chromosome missegregation and aneuploidy, a hallmark of many cancers [3].

To ensure chromosome bipolar attachment, active SAC prevents anaphase entry, allowing more time for the correction of chromosome attachment errors. Furthermore, the destabilization of kinetochore-microtubule (KT-MT) interaction enables error correction. The stability of KT-MT interaction relies on a delicate balance between kinase and phosphatase activity at kinetochores. In general, phosphorylation of some kinetochore substrates by a conserved kinase Aurora B (Ipl1 in budding yeast) destabilizes KT-MT interaction to facilitate error correction. This phosphorylation is reversed by increasing phosphatase activity at kinetochores to stabilize the KT-MT interaction [4]. Once bipolar attachment has been established, the tension on sister kinetochores is believed to silence the SAC to allow anaphase entry [5]. In addition, tightly regulated protein phosphatase activity at kinetochores is also critical to accurate chromosome segregation during meiosis.

If any of these mechanisms fail, the consequence is chromosome missegregation or aneuploidy, which is a hallmark of cancer cells and is the cause for genetic disorders like Down syndrome (Trisomy 21). The regulation of simultaneous kinase and phosphatase activity at the kinetochore remains one of the most important yet elusive undertakings. Further unveiling of these detailed mechanisms will elucidate additional key players in chromosome bipolar attachment and prove beneficial for cancer drug development. In this review, we discuss the antagonistic relationships between kinases and phosphatases and their roles in promoting accurate chromosome segregation, with a focus on budding yeast *Saccharomyces cerevisiae*.

## 2. Kinases Involved in the Establishment of Chromosome Bipolar Attachment

### 2.1. Aurora B/Ipl1 Kinase

Several kinases are critical for chromosome bipolar attachment, including Aurora B (Ipl1 in budding yeast), cyclin-dependent kinase (CDK), Plk1 (Cdc5 in budding yeast), and a conserved kinase, Mps1. In mammalian cells, the chromosomal passenger complex (CPC) is made up of Aurora B and three other proteins, INCENP, Survivin, and Borealin [6]. The CPC in budding yeast consists of Ip11 kinase and three other proteins, Sli15, Bir1, and Nbl1 [7,8]. Before anaphase entry, kinetochore/centromere localized Aurora B/Ipl1 kinase phosphorylates some kinetochore proteins to facilitate chromosome bipolar attachment by destabilizing KT-MT interaction. During anaphase, the CPC relocates from the kinetochore to the spindle to stabilize the spindle midzone [9].

The function of Ipl1 kinase in kinetochore attachment was demonstrated by examining the stability of KT-MT interaction in yeast cells lacking DNA replication. In wild-type cells lacking DNA replication, the destabilized KT-MT interaction allows kinetochore detachment and reattachment, resulting in the random distribution of the unduplicated 16 chromosomes to the two daughter cells. However, these chromosomes only move into the daughter cells with the old spindle poles in *ipl1* mutants, indicating that kinetochore attachment to microtubules from the parent spindle pole is not disrupted [10]. This is the first in vivo evidence for the function of Ipl1 kinase in destabilizing KT-MT interaction in budding yeast.

Further studies suggest that Ipl1 regulates the stability of KT-MT interaction by phosphorylating Dam1, one of the ten subunits in the Dam1 (DASH) complex [11,12]. This complex forms a ring structure around microtubules [13,14], and mediates KT-MT interaction by binding to an outer kinetochore complex, Ndc80 [15,16]. Ipl1 phosphorylates Dam1 protein on four consensus sites: S20, S257, S265, and S292. Mutations of all four Ipl1 consensus sites in Dam1 protein to alanine causes lethality, but mutations in three sites (S257, S265, S292) generate a viable phospho-deficient *dam1-3A* mutant [12]. The presence of a bacterially-expressed Dam1 complex significantly enhances the association of the Ndc80 complex with microtubules in vitro, indicating that the Dam1 complex recruits the Ndc80 complex onto microtubules. Strikingly, phosphorylation of the Dam1 complex by Ipl1 kinase abolished the association of the Ndc80 complex with microtubules. The phosphorylation of S20 at the N-terminus of Dam1 decreases the microtubule affinity of the Dam1 complex, but does not affect microtubule recruitment of the Ndc80 complex [17,18]. These results suggest that Ipl1-dependent phosphorylation of Dam1 compromises the association of the Dam1 complex with both microtubules and the Ndc80 complex. This mechanism seems conserved in mammalian cells, as Aurora B kinase phosphorylates some subunits in the Ska complex, the functional orthologue of the Dam1 complex, to destabilize the interaction of the Ska complex with other kinetochore proteins [19]. In addition, Ipl1 kinase phosphorylates centromeric histone H3-like protein Cse4 in budding yeast, and this phosphorylation appears to destabilize kinetochore attachment as well [20].

The Ndc80 complex localizes at the outer kinetochore to mediate KT-MT interactions. In addition to Dam1, Ndc80 protein is also an Ipl1 substrate [12,21]. Seven Ipl1 consensus sites have been identified in vivo on the unstructured N-terminal domain of Ndc80, including T21, S37, T54, T71, T74, S95, and S100. Mutation of these phosphorylation sites on Ndc80 (*ndc80-7A*) increases sensitivity to the microtubule depolymerizing agent benomyl. However, *ndc80-7A* mutant cells exhibited intact SAC function in response to spindle disruption [21]. Therefore, Ndc80 phosphorylation likely contributes to chromosome bipolar attachment especially after KT-MT interaction is disrupted by spindle poisons, such as benomyl and nocodazole. In mammalian cells, Ndc80/Hec1 is also a substrate of Aurora B kinase and Ndc80 dephosphorylation promotes kinetochore-microtubule attachments [22]. More work is needed to understand the function of Ndc80 phosphorylation in chromosome bipolar attachment in budding yeast.

Previous studies also suggest the role of Ipl1 kinase in checkpoint response to faulty chromosome attachment. *ipl1* mutants enter anaphase in the presence of tensionless chromosome attachment, but show intact checkpoint arrest in response to kinetochore detachment caused by spindle disruption [23]. One possibility is that functional Ipl1 destabilizes KT-MT interaction to generate detached kinetochores for SAC activation [24]. Alternatively, Ipl1-dependent phosphorylation prevents SAC silencing. Our lab found that disruption of the Cik1/Kar3 motor complex induces tensionless syntelic attachment, where sister kinetochores are attached to microtubules from the same spindle pole. Intact SAC and functional Ipl1 are required for the anaphase delay induced by syntelic attachment [25]. Therefore, Ipl1 kinase not only promotes error correction for kinetochore attachment by destabilizing the attachment, but also delays anaphase onset in response to tensionless chromosome attachment.

### 2.2. Cyclin-Dependent Kinase (CDK)

CDK kinase activity is important throughout the cell cycle. Sli15, a budding yeast CPC component, is a CDK substrate. Sli15 phosphorylation is required for kinetochore/centromere localization of CPC, while its dephosphorylation by Cdc14 phosphatase triggers its translocation to the spindle [26]. Therefore, CDK activity may promote chromosome bipolar attachment indirectly by facilitating CPC localization at kinetochores. In fission yeast *Schizosaccharomyces pombe,* CDK phosphorylates CPC component Bir1/Survivin to promote chromosome biorientation by targeting CPC to the kinetochore/centromeres, while phospho-deficient forms of Bir1 have defects in biorientation. In non-phosphorylatable *bir1-8A* mutant cells, centromeric CPC localization is abolished, and a high incidence of lagging chromosomes was observed during anaphase [27]. Surprisingly, a previous study shows that Bir1 is dispensible for kinetochore/centromere localization of CPC in budding yeast [28]. Similarly, recent evidence demonstrates that the direct interaction of the Ipl1-Sli15 complex with kinetochore proteins is independent of Bir1 [29]. Regardless, CDK-dependent phosphorylation of CPC and kinetochore/centromere localization promote the establishment of chromosome bipolar attachment.

### 2.3. Mps1 Kinase

Mps1 (monopolar spindle) kinase is evolutionarily conserved across species and its role in SAC activation is well established. In budding yeast, *mps1* mutants fail to arrest the cell cycle in response to spindle disruption. Mps1 overexpression constitutively activates the SAC, and this activation persists regardless of KT-MT interaction [30,31]. Kinetochore-localized Mps1 phosphorylates the MELT motifs of the kinetochore protein Spc105. The MELT reading domain in SAC protein Bub3 then recruits the Bub1/Bub3 complex to the kinetochore [32,33]. In addition, Mps1-dependent phosphorylation of Bub1 further promotes the recruitment of Mad1 to kinetochores for SAC activation [34]. This Mps1-dependent phosphorylation of kinetochore protein Spc105 (Knl1 in mammalian cells) and SAC protein Bub1 appears conserved from yeast to human cells [35,36,37]. In mammalian cells, the phosphorylation of Mad1 by Mps1 promotes its association with Cdc20, the regulatory subunit of the anaphase promoting complex APC/C [38]. Therefore, Mps1 phosphorylates several kinetochore proteins essential for SAC activation.

In addition to its role in SAC function, Mps1 is also important for chromosome biorientation. Mps1 promotes reorientation of kinetochore-spindle pole connections and eliminates tensionless attachment. Interestingly, the function of Mps1 in chromosome bipolar attachment is likely independent of Ipl1 [39]. Mps1 phosphorylates Dam1 on six serines in vitro and the phosphorylation of at least two of these sites (S218 and S221) is required for association of kinetochores to microtubule plus ends. Shockingly, cells with S218 and S221 mutations segregate their chromosomes properly [40]. Therefore, the exact mechanism of how Mps1 kinase activity contributes to biorientation remains to be determined.

### 2.4. Cdc5 Kinase

Budding yeast Cdc5 is a conserved protein kinase and its human homologue is Plk1. Studies in budding yeast have established that Cdc5 kinase promotes mitotic exit [41,42,43]. In addition, Cdc5 localizes at centromeres and cohesin-associated regions to facilitate sister chromatid separation [44,45]. Cdc5 kinase phosphorylates cohesin Scc1/Mcd1 to support its cleavage by separase [46,47]. Moreover, hyperactive Cdc5 removes cohesin regulator Pds5 from chromosomes to assist chromosome resolution [48]. A recent study from the Basrai lab indicates that the centromeric histone H3 variant Cse4 interacts with Cdc5 to enable centromere recruitment of Cdc5. In turn, Cdc5 phosphorylates Cse4, and this phosphorylation further stimulates centromere recruitment of inner kinetochore protein Mif2 as well as cohesin [49]. Therefore, Cdc5-mediated Cse4 phosphorylation helps kinetochore assembly and cohesion establishment at centromeres. Moreover, active Cdc5 facilitates sister chromatid separation. Recent advances about the role of Cdc5 kinase in faithful chromosome segregation are reviewed by Mishra and Basrai [50].

In summary, Ipl1, CDK, Mps1, and Cdc5 kinases work together to promote chromosome bipolar attachment by facilitating the correction of erroneous KT-MT interaction and activating the checkpoint to prevent anaphase onset in the presence of erroneous attachment. Once cells have established chromosome bipolar attachment, these phosphorylation events are reversed by protein phosphatases to silence the SAC and stabilize KT-MT interaction for chromosome segregation. Dephosphorylation events are spatiotemporally regulated to ensure faithful chromosome segregation, and several phosphatases are involved in this process. The association of phosphatases with different regulators confers their subcellular localization [5,51,52]. In the next Section, we cover the major phosphatases in budding yeast involved in the establishment of chromosome biorientation with an emphasis on mitosis.

## 3. Protein Phosphatases Involved in Chromosome Bipolar Attachment

### 3.1. Protein Phosphatase 1 (PP1) and KT-MT Attachment

Aurora B/Ipl1-mediated protein phosphorylation at kinetochores destabilizes their interaction with microtubules to facilitate error correction. Once bipolar attachment is established, this phosphorylation needs to be reversed to stabilize KT-MT interaction for efficient chromosome segregation in anaphase. The phosphorylation of Dam1 fluctuates during the cell cycle and a dramatic dephosphorylation was observed at a narrow window, likely prior to anaphase onset, indicating cell cycle-regulated Dam1 dephosphorylation [11,53]. This phosphorylation persists in cells that lack tension, which likely destabilizes KT-MT interaction to promote error correction [54]. PP1 has been shown to reverse the phosphorylation imposed by Aurora B/Ipl1 in both yeast and higher eukaryotes [51,55]. Five genes in mammals encode the catalytic subunit of PP1, whereas one single gene, *GLC7*, does the same in budding yeast [56,57]. The *GLC7* gene is essential, and enhanced Dam1 phosphorylation was detected in a conditional *glc7* mutant allele, indicating the role of PP1 in Dam1 dephosphorylation in vivo [58]. In addition, mutations in *GLC7* genes suppress the temperature sensitivity of *ipl1* mutants, further supporting the notion that PP1 reverses the phosphorylation imposed by Ipl1 kinase [59,60,61].

The role of Dam1 dephosphorylation in KT-MT interaction has been studied using phospho-deficient *dam1-3A* mutants, where three Ipl1 consensus sites are mutated to alanine [12]. The stability of KT-MT interaction in *dam1-3A* cells lacking DNA replication was examined as described by the Nasmyth lab [10]. A strongly enhanced attachment of unduplicated chromosomes with the old spindle poles was observed in *dam1-3A* mutant cells, suggesting that the reversion of Ipl1-dependent Dam1 phosphorylation stabilizes KT-MT interaction in vivo. Moreover, *dam1-3A* cells exhibit sensitivity to spindle poison nocodazole, and high rates of chromosome missegregation were detected in *dam1-3A* after release from nocodaozle treatment. This phenotype is likely due to the failure of error correction caused by stable KT-MT interaction [53]. Hyper-phosphorylation of Dam1 protein was observed in *glc7* mutant cells [58], suggesting that PP1 dephosphorylates Dam1. Conditional alleles of *glc7* cause cell cycle arrest at metaphase, and this arrest is abolished by SAC deficiency, indicating a defect in SAC silencing and/or chromosome attachment in this mutant [62,63]. Hyper-phosphorylation of kinetochore protein Ndc10 was observed in *glc7-10* mutant, but the function of Ndc10 phosphorylation in kinetochore attachment remains unclear. Unlike Dam1 and Ndc10, the role of PP1 in the reversion of Ipl1-dependent Ndc80 phosphorylation is less clear. It will be interesting to understand how this phosphorylation contributes to chromosome bipolar attachment (Figure 1).

### 3.2. PP1 and Checkpoint Regulation

*Glc7* overexpression bypasses cell cycle arrest induced by spindle disruption or tensionless attachments [64], indicating that PP1 is involved in both chromosome biorientation and SAC silencing. As discussed previously, Mps1 phosphorylates the MELT motifs of kinetochore protein Spc105, which sequentially recruits the Bub1/Bub3 and Mad1/Mad2 SAC proteins for SAC activation. Mutating six phosphorylated threonines to alanine generates a *spc105-6A* mutant that abolishes Bub1/Bub3 binding and exhibits a defective SAC [32]. Interestingly, Spc105 has a conserved PP1 binding motif, and mutation of this PP1 binding motif leads to SAC-dependent cell cycle arrest [65]. A reasonable explanation is that the association of PP1 with Spc105 reverses Mps1-dependent Spc105 phosphorylation and silences the SAC.

In budding yeast, Mps1 directly interacts with and phosphorylates the N-terminal of the kinetochore protein Ndc80. Strains in which the phosphorylation residues were mutated to alanine fail to show cell cycle arrest when treated with nocodazole. Alternatively, phospho-mimetic mutants show constitutive metaphase arrest, but this arrest is eliminated when combined with SAC mutants [66]. It remains to be determined, then, how Ndc80 phosphorylation regulates checkpoint activity. Moreover, it is unclear if PP1 reverses the phosphorylation of Ndc80 imposed by Mps1 to regulate SAC activity.

Ipl1 kinase is required to prevent anaphase onset in response to tensionless attachment, and Dam1 is a substrate of both Ipl1 kinase and PP1. Therefore, the role of Dam1 dephosphorylation in the checkpoint response to syntelic attachment was examined. Interestingly, phospho-deficient *dam1-3A* mutants show no cell cycle delay when syntelic attachment is induced, resulting in chromosome missegregation and viability loss. In clear contrast, phospho-mimetic *dam1-3D* mutants exhibited significant delay in anaphase entry, and this delay is abolished by the loss of function of SAC without causing dramatic viability loss. This result indicates that Dam1 phosphorylation delays anaphase entry, and chromosome detachment may not be the only cause for this delay. We speculate that the dephosphorylation of Dam1 by PP1 silences the SAC to allow anaphase entry [67]. Therefore, the results from *ipl1* and phospho-deficient *dam1-3A* mutants are consistent and support the conclusion that Dam1 dephosphorylation promotes anaphase entry. Further studies are required to understand how PP1 activity at kinetochores is regulated in a timely manner for Dam1 dephosphorylation. Another important open question is how Dam1 dephosphorylation negatively regulates SAC activity (Figure 2).

### 3.3. Kinetochore Recruiters for PP1

Considering the critical role of PP1 in chromosome bipolar attachment and SAC silencing, its activity needs to be tightly regulated in a spatiotemporal manner. In budding yeast, three proteins are known to be involved in PP1 kinetochore localization: kinetochore protein Spc105, kinesin-5 motor protein Cin8, and another kinetochore protein Fin1. Spc105 harbors a conserved PP1 binding motif RVxF, and mutations in this motif (*spc105-*RASA) are lethal to yeast cells. However, this lethality is suppressed by deleting checkpoint protein Mad2, indicating the requirement for PP1-Spc105 interaction for SAC silencing. More evidence suggests that the cell cycle arrest in *spc105*-RASA is not due to chromosome biorientation defect [65]. The detailed mechanism of how PP1 cooperates with Spc105 to silence the checkpoint has yet to be resolved, but it is possible that Spc105-PP1 reverses Mps1 and/or Ipl1-dependent phosphorylation for SAC silencing.

A recent study in budding yeast indicates that the bidirectional motor protein Cin8 is another PP1 regulator. Cin8 belongs to the kinesin-5 family and assists in spindle orientation and force generation through anaphase [68]. Loss of Cin8 function delays cell cycle progression, presumably due to a spindle defect or failure of tension generation [69,70,71]. Cin8 localizes to the kinetochore during metaphase, and this localization depends on the Ndc80 complex, kinetochore microtubules, and the Dam1 complex [72]. The Cin8 C-terminus interacts with PP1 through a conserved PP1-binding motif (RVKW), but Cin8-PP1 interaction is not necessary for Cin8 kinetochore localization. Experiments utilizing a FRET tension biosensor show that Cin8-PP1 binding mutants (*cin8-KAAKA*) accumulate in metaphase with tensionless Ndc80-microtubule attachments [72]. Interestingly, increased Dam1 phosphorylation was detected in *cin8* mutant cells in a recent study (Mukherjee et al., 2019). Further studies are required to verify if kinetochore-localized Cin8-PP1 dephosphorylates Dam1 to stabilize KT-MT interaction and silence the SAC. Because *cin8∆* cells are synthetically lethal with SAC mutants, Cin8 likely plays a role in the establishment of correct kinetochore attachment [73]. Therefore, the function of Cin8 in kinetochore attachment and PP1 kinetochore recruitment might be separable. The target(s) of Cin8-PP1 at the kinetochore and the function of this dephosphorylation need to be further explored.

Fin1 is another identified PP1 regulatory subunit. Budding yeast cells undergo a closed mitosis, meaning the nuclear envelope does not breakdown during cell division. Fin1 protein localizes in the nucleus through metaphase, but moves to the spindle in anaphase to promote spindle stability. The phosphorylation of Fin1 by S-phase CDK prevents Fin1 translocation to the kinetochore and spindle [74,75]. In addition, Fin1 forms a complex with PP1, and phosphorylated Fin1 associates with a 14-3-3 protein Bmh1 to prevent kinetochore localization of Fin1-PP1. Fin1 dephosphorylation by Cdc14 phosphatase during early anaphase enables kinetochore recruitment of Fin1-PP1 [76]. Although the exact target of Fin1-PP1 is unclear, we recently found that Fin1-PP1 kinetochore localization is necessary to remove SAC protein Bub1 from the kinetochore, because *fin1∆* cells show persistent Bub1 kinetochore localization in anaphase [77]. Mutating the CDK phosphorylation sites of Fin1 to alanine produces a non-phosphorylatable Fin1-5A protein that is prematurely targeted to the kinetochore. This mutant shows increased sensitivity to syntelic attachments, indicating premature SAC silencing in response to tensionless attachments. Intriguingly, *fin1∆* mutants do not show noticeable anaphase delay, thus the significance of Fin1-PP1 dependent kinetochore removal of SAC protein Bub1 remains to be determined.

### 3.4. Phosphatase Cdc14

Cdc14 phosphatase specifically dephosphorylates CDK substrates [78]. Before anaphase entry, Cdc14 is sequestered in the nucleolus by association with a nucleolar protein Net1/Cfi1 [79,80]. After anaphase entry, nucleolar localized Cdc14 is released sequentially by the activation of two mitotic pathways: FEAR and MEN [51]. The activation of the Cdc14 Early Anaphase Release (FEAR) pathway promotes a transient Cdc14 release during early anaphase [43]. During the metaphase-anaphase transition, degradation of the securin Pds1 frees separase (Esp1), which promotes the dissociation of phosphatase PP2A^Cdc55^ from Net1. This allows Net1 phosphorylation by CDK and triggers the subsequent Cdc14 release from the nucleolus [81,82]. FEAR-mediated Cdc14 release reverses the phosphorylation imposed by S-phase CDK. In late anaphase, a robust Cdc14 release is triggered by the Mitotic Exit Network (MEN), resulting in the dephosphorylation of all CDK substrates [83,84].

As discussed previously, CPC regulates the stability of KT-MT interaction and the spindle midzone [27]. Prior to anaphase, CDK phosphorylates one CPC component, Si15/INCENP, to prevent CPC translocation from the kinetochore to the spindle midzone [85]. This mechanism enables Ipl1-dependent destabilization of KT-MT interaction and error correction prior to anaphase onset. Once biorientation triggers SAC silencing, the release of Cdc14 during early anaphase dephosphorylates Sli15 to promote CPC translocation from the kinetochore to the anaphase spindle [26]. Mutating the CDK phosphorylation sites on Sli15 generates a phospho-deficient mutant *sli15-6A*, which shows premature CPC translocation from the kinetochore. Consistently, elimination of Cdc14 phosphatase (*cdc14-td*) results in persistent Sli15 phosphorylation and kinetochore localization [26]. Therefore, Cdc14 may indirectly regulate chromosome biorientation by triggering CPC translocation. As *sli15-6A* mutant exhibits premature CPC translocation, it is worthwhile to examine the effect on the correction of kinetochore attachment errors. The result will clarify if premature CPC translocation compromises the error correction process.

In addition to Sli15, Cdc14 dephosphorylates kinetochore protein Fin1 [75]. As discussed above, Fin1 phosphorylation by S-phase Cdk1 prevents kinetochore localization of Fin1-PP1 [76]. FEAR-mediated Cdc14 release leads to Fin1 dephosphorylation and kinetochore recruitment of Fin1-PP1, which clears SAC protein Bub1 from the kinetochore. The results from phospho-deficient *fin1* mutants (*fin1-5A*) support the expected consequence of Fin1 dephosphorylation [77,86]. Therefore, Cdc14 phosphatase also regulates Ipl1/PP1 balance at the kinetochore through Fin1 dephosphorylation during anaphase, although the biological significance of this regulation is not fully understood yet (Figure 3).

### 3.5. Protein Phosphatase 2A (PP2A)

PP2A also plays a role in sister kinetochore biorientation. PP2A is a trimeric serine/threonine phosphatase composed of one catalytic subunit (C), one scaffold subunit (A), and one regulatory subunit (B), which confers substrate specificity [87]. Rts1 and Cdc55 are the two PP2A regulatory subunits in budding yeast. PP2A^Rts1^ cooperates with a centromeric protein, Shugoshin (Sgo1), to promote chromosome biorientation. Sgo1 was originally discovered from both budding and fission yeast because it protects centromeric cohesin from premature cleavage by separase during meiosis I [88,89]. Sgo1 recruits PP2A^Rts1^ to centromeres to dephosphorylate cohesin, thereby inhibiting its cleavage by separase until anaphase II during meiosis. In mitosis, PP2A^Rts1^ is also recruited to centromeres, and this recruitment is dependent on the kinase activity of SAC protein Bub1. Kinetochore localized Bub1 phosphorylates histone H2A adjacent to kinetochores. Previous studies indicate that phosphorylated H2A recruits Sgo1 to the centromere, and Sgo1 further recruits PP2A^Rts1^ [90,91]. However, recent evidence suggests that centromere H3 variant Cse4 interacts with the N-terminal domain of Sgo1. This interaction is responsible for Sgo1 recruitment to centromeres, but not to pericentromere regions [92,93]. Therefore, two distinct mechanisms could contribute to the association of Sgo1 with chromatin.

In budding yeast, Sgo1-PP2A^Rts1^ is critical to ensure that sister kinetochores are intrinsically biased toward capture by microtubules from opposite poles [94]. Consistently, elimination of Bub1 kinase activity (*bub1∆K*) abolishes Sgo1 centromere localization and leads to high levels of chromosome missegregation, although the SAC activity is intact in response to spindle poison nocodazole [90]. The function of Sgo1 in chromosome biorientation depends on its recruitment of PP2A^Rts1^ to the centromere [95]. Further evidence shows that Sgo1, together with PP2A^Rts1^, ensures localization of condensin to the pericentromeric chromatin in yeast cells. This likely contributes to accurate conformation of the pericentric region for bipolar attachment formation. In addition, Sgo1 maintains Ipl1 at kinetochore/centromere regions prior to tension generation [96,97]. Thus, Sgo1 plays a dual role in promoting chromosome biorientation in budding yeast. First, Sgo1 works with PP2A^Rts1^ to recruit condensin to the pericentric region, which modulates the conformation of this region to facilitate biorientation. Second, Sgo1 maintains Ipl1 at the kinetochore to facilitate error correction. Once tension generates from biorientation, Sgo1 is removed from pericentric regions by proteasome-mediated Sgo1 degradation through E3 ligase APC/C [95,98].

Although *sgo1∆* cells show intact SAC function in response to spindle disruption, the mutant cells fail to arrest the cell cycle in response to tension defects [99]. Our recent results show that deletion of the Bub1 kinase domain (*bub1-ΔK*) also leads to checkpoint deficiency in response to tensionless chromosome attachment, but the checkpoint response to spindle disruption is intact. Moreover, mutation of the Bub1 kinase phosphorylation sites at H2A, deletion of PP2A regulatory subunit *RTS1*, or abolishment of Sgo1-PP2A^Rts1^ interaction also compromises checkpoint arrest in response to tensionless attachment. Therefore, in budding yeast, the Bub1-H2A-Sgo1-PP2A^Rts1^ axis prevents SAC silencing prior to chromosome bipolar attachment [100]. Taken together, these results suggest that PP2A^Rts1^ promotes chromosome bipolar attachment by ensuring the correct conformation of pericentric chromatin and by preventing anaphase onset prior to chromosome biorientation and tension generation.

In budding yeast, Cdc55 is another B regulatory subunit of PP2A. PP2A^Cdc55^ may regulate chromosome bipolar attachment indirectly by preventing early Cdc14 release from the nucleolus and untimely cleavage of sister chromatid cohesin. PP2A^Cdc55^ dephosphorylates nucleolar protein Net1 to enable its sequestration of Cdc14 within the nucleolus until early anaphase [81,82,101]. *cdc55∆* mutant cells show premature release of Cdc14, which allows the dephosphorylating of CDK substrates, including the CPC component Sli15. We found that Cdc55 is required for efficient cell cycle delay in response to tensionless attachment [77]. It will be interesting to examine the Ipl1 kinetochore localization in *cdc55* mutant cells at metaphase to clarify if the dysfunction of PP2A^Cdc55^ compromises CPC kinetochore localization. In budding yeast, the phosphorylation of sister chromatid cohesin Scc1 by polo-like kinase Cdc5 facilitates Scc1 cleavage by separase Esp1 [46]. In contrast, PP2A^Cdc55^ dephosphorylates Scc1 and prevents its cleavage. Consistently, *cdc55*∆ mutant cells exhibit premature cohesin cleavage, which likely impairs chromosome bipolar attachment [102,103].

In mammalian cells, Sgo1 also works with PP2A^B56^ (B56 is the Rts1 yeast homolog) to protect centromeric cohesion [104]. Kinetochore-localized Sgo1-PP2A protects cohesin from cleavage by dephosphorylating the cohesion regulator sororin, a CDK substrate [105]. Sgo1 also recruits Aurora B to the kinetochore. Depletion of PP2A^B56^ increases phosphorylation of some Aurora B substrates and destabilizes KT-MT interaction, indicating that PP2A^B56^ antagonizes the phosphorylation imposed by Aurora B at the kinetochore [106,107]. This data supports the role of PP2A^B56^ in establishing chromosome biorientation by protecting centromeric cohesion. PP2A^B56^ also antagonizes the phosphorylation imposed by Aurora B kinase at kinetochores, but it will be important to understand how these PP1 activities at kinetochores are regulated spatiotemporally. 

## 4. The Function of Phosphatases in Meiosis

Similar to mitosis, accurate chromosome segregation depends on highly regulated and time-sensitive kinase/phosphatase activity. During meiosis, diploid cells undergo two distinct rounds of chromosome segregation after only one round of DNA replication. In meiosis I, pairs of homologous chromosomes segregate when cohesion still connects the sister chromatids at the centromeric regions. This process requires sister kinetochores to be in a mono-oriented state [108]. In meiosis II, the sister chromatids are separated to opposite spindle poles, much like they are in mitosis. A meiosis-specific mechanism prevents biorientation of kinetochores during meiosis I, and in budding yeast, the four-subunit monopolin complex (Mam1, Csm1, Lrs4, and Hrr25) confers this responsibility [109,110,111]. Monopolin associates with sister kinetochores, and the V-shaped confirmation of the Csm1-Lrs4 complex facilitates this association [112]. Therefore, the monopolin-kinetochore association is the basis for the establishment of monopolar attachment of sister kinetochores at meiosis I.

Ipl1 localizes to the kinetochore during metaphase I and II in meiosis and *ipl1* mutants show high levels of missegregation for both homologs and sister chromatids [113]. Ipl1 was also found to release KT-MT associations after meiotic entry, liberating chromosomes for homologous pairing. Moreover, Ipl1 releases improper connections between chromosome pairs and microtubules [114]. Therefore, Ipl1 likely destabilizes KT-MT interactions in both mitosis and meiosis to ensure correct kinetochore attachments.

Faithful meiotic chromosome segregation also depends on a tightly regulated balance of kinase/phosphatase activity at kinetochores during the metaphase to anaphase transition. Sister chromatids separate during anaphase II in meiosis, thus there must be a mechanism to prevent sister chromatid separation until this time. In budding and fission yeast, sister chromatid cohesion during meiosis depends on a meiosis-specific cohesin subunit Rec8, the equivalent of mitotic cohesin Sccl [115,116]. Two rounds of Rec8 cleavage occur during meiosis. The first round happens on chromosome arms at the onset of anaphase I, resulting in dyads where sister chromatids are connected at their centromeres by cohesion. At the onset of anaphase II, the second round Rec8 cleavage leads to the disjunction of sister chromatids, generating the single-copy genome. In yeast cells, two kinases and one phosphatase play a key role in the stepwise cleavage of Rec8. Casein kinase Hrr25 and Dbf4-Cdc7 kinases phosphorylate Rec8 to prime its cleavage by separase [117,118,119]. In contrast, centromere-localized Sgo1 prevents Rec8 from cleavage [88,120,121]. Sgo1 recruits PP2A^Rts1^ to centromeres to dephosphorylate Rec8 and protect centromeric cohesion during meiosis [89,118]. Similar to mitosis, the centromeric localization of Sgo1 during meiosis also depends on Bub1-mediated phosphorylation of H2A [91]. The activation of ubiquitin-ligase APC/C-Cdc20 removes PP2A^Rts1^ from centromeres by targeting Sgo1 for degradation, which leads to Rec8 phosphorylation and cleavage in meiotic anaphase II [122]. In addition, Sgo1 recruits Ipl1 to kinetochores during meiosis, which maintains PP2A^Rts1^ activity at centromeres [123]. Therefore, Ipl1 may play a key role in meiotic cohesion maintenance between sister chromatids. It will be interesting to understand how Ipl1 regulates centromeric localization of PP2A^Rts1^ at the molecular level.

## 5. Conclusions

Faithful chromosome segregation is indispensable to all cell life and therefore must be under tight control. The establishment of bipolar attachment is key to faithful chromosome segregation during mitosis. Extensive studies have unveiled several mechanisms to ensure chromosome bipolar attachment, and much of this control relies on an intricate balance between kinase and phosphatase activity at the kinetochore. First, centromere localized Sgo1-PP2A^Rts1^ in budding yeast ensures a proper sister kinetochore conformation that is biased toward biorientation, and Sgo1-PP2A^Rts1^-dependent recruitment of condensin at the centromeric region likely plays a critical role for this mechanism. Second, kinetochore/centromere localized Ipl1 kinase phosphorylates some kinetochore proteins to destabilize KT-MT interaction and assist in error correction. The recruitment of PP1 to kinetochores reverses this phosphorylation and stabilizes KT-MT interaction. Third, the kinetochore serves as a hub to sense faulty kinetochore attachment and activate the SAC to prevent anaphase entry. SAC activation sets aside more time for cells to correct kinetochore attachment errors.

Two layers of SAC regulation have been uncovered, and the first layer relies on kinetochore-localized Mps1 kinase. Mps1 phosphorylates Spc105 to enable SAC activation at the kinetochore, but the recruitment of PP1 to kinetochores silences the SAC to trigger anaphase entry. The second layer of SAC regulation depends on Ipl1-mediated phosphorylation of kinetochore proteins. In response to tensionless kinetochore attachment, Ipl1-dependent phosphorylation of a kinetochore protein Dam1 delays anaphase onset, likely by preventing SAC silencing. In contrast, dephosphorylation of Dam1 by PP1 allows anaphase entry in the presence of tensionless attachments. Moreover, the Bub1-Sgo1-PP2A^Rts1^ axis is also required to delay anaphase entry in response to tensionless attachment. It is clear that protein kinases (Ipl1 and Mps1) and protein phosphatases (PP1 and PP2A) directly regulate chromosome bipolar attachment. CDK kinase and Cdc14 phosphatase may regulate this process indirectly by modulating the Ipl1/PP1 balance at the kinetochore.

During meiosis, homologous chromosomes segregation occurs, while centromere-localized Sgo1-PP2A^Rts1^ restrains sister chromatid separation during meiosis I by protecting centromeric cohesin from cleavage. In meiosis, the removal of Sgo1-PP2A^Rts1^ from centromeres allows sister chromatid segregation. Failure in any of these mechanisms will result in chromosome missegregation or aneuploidy, which contributes to cancer development and birth defects. Further understanding of the detailed mechanisms that ensure chromosome bipolar attachment may provide new strategies for cancer prevention or treatment (Figure 4).

## Figures and Tables

**Figure 1 ijms-20-06182-f001:**
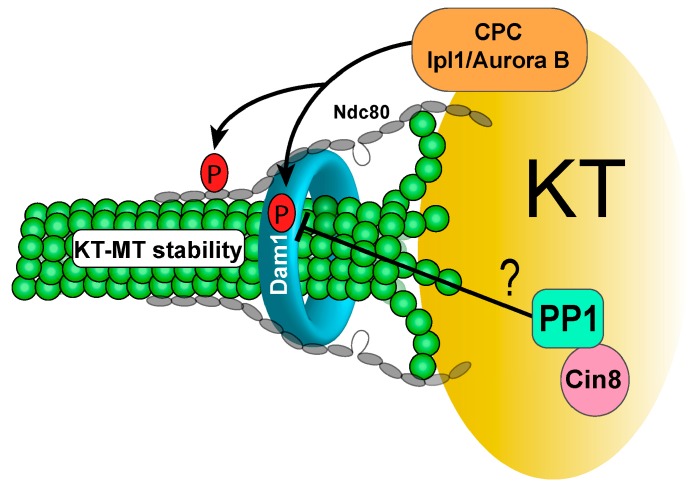
PP1 antagonizes Ipl1/Aurora B-dependent Dam1 phosphorylation, possibly through the PP1-regulator Cin8. Dam1 dephosphorylation stabilizes the KT-MT interaction at anaphase onset. (KT: kinetochore, MT: microtubule, CPC: chromosome passenger complex). Black arrows indicate positive regulation, and the T arrow indicates inhibition. The question mark indicates unverified regulation.

**Figure 2 ijms-20-06182-f002:**
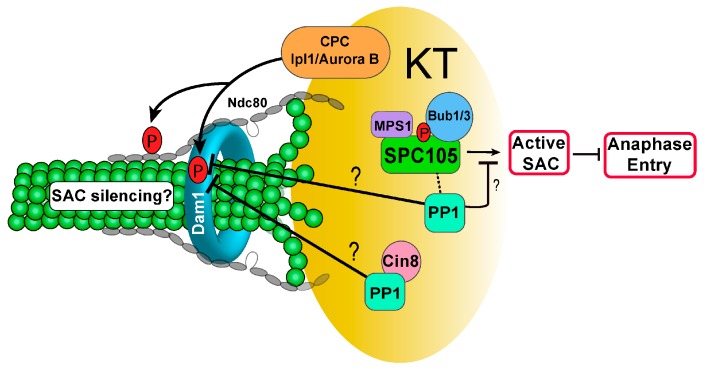
Spc105-associated PP1 is required to silence the spindle assembly checkpoint (SAC) through an unknown mechanism. In addition to KT stability, Cin8-PP1 may be involved in checkpoint silencing via Dam1 dephosphorylation. A black arrow indicates positive regulation, but a T arrow indicates inhibition. Question marks mean unverified regulation.

**Figure 3 ijms-20-06182-f003:**
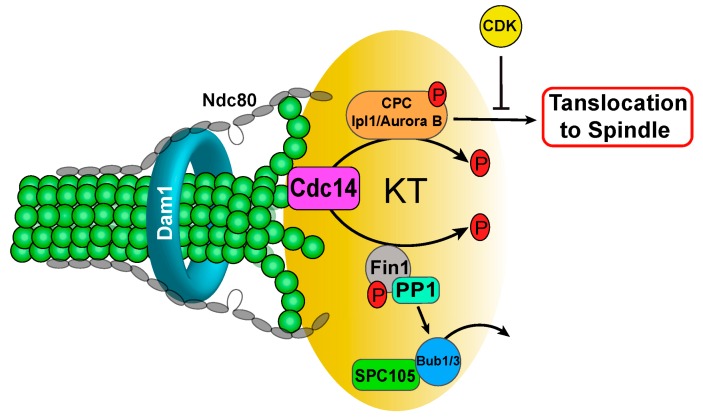
FEAR-activated Cdc14 dephosphorylates chromosomal passenger complex (CPC) component Sli15 to promote its relocation to the spindle during anaphase. Cdc14-dependent Fin1 dephosphorylation allows Fin1-PP1 KT localization to promote the dissociation of SAC proteins Bub1/3 from the KT during anaphase. A black arrow indicates positive regulation, but a T arrow indicates inhibition.

**Figure 4 ijms-20-06182-f004:**
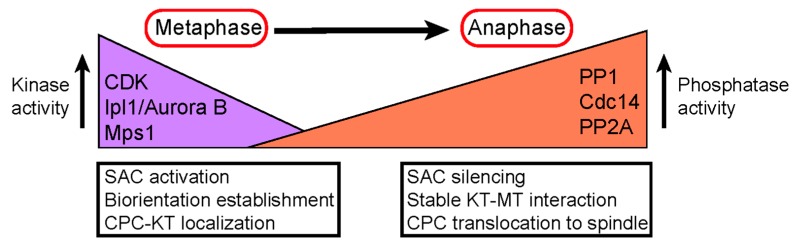
Model describing the change in kinase/phosphatase balance at the kinetochore during the metaphase to anaphase transition. Maintaining this balance is essential for faithful chromosome segregation. Listed in the purple block are the major kinases, with the black arrow indicating that kinase activity is high. Listed in the orange block are the major phosphatases, with the black arrow indicating that phosphatase activity is high. The arrow between metaphase and anaphase indicates the cell cycle stage transition.

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
