# Peer review of "The Opposing Functions of Protein Kinases and Phosphatases in Chromosome Bipolar Attachment"

_ijms, 2019, doi:10.3390/ijms20246182_

Round 1

Reviewer 1 Report

Title: The function and regulation of protein phosphatases in establishing chromosome bipolar attachment.

Authors: Delaney Sherwin and Yanchang Wang

In this manuscript, authors have reviewed the current state of literature on the role of protein phosphatases in regulation of chromosome bipolarity for the maintenance of chromosome stability. The objectives of this review are logical, timely, thought-provoking, and provide deeper insights into the current state of knowledge in the field. The manuscript is very well written and organized into sections that are easy to follow and understand for the broad readerships of the journal. Overall, manuscript is acceptable pending minor revisions outlined below.

Specific Comments:

Authors have very elegantly discussed the role of different kinases (for example Ipl1, Mps1, Cdk), but have omitted the role of polo-like kinase Cdc5, which also plays a critical role in kinetochore function. This review appears incomplete without information on Cdc5. Authors should add some information on this topic from recent papers on Cdc5-mediated phosphorylation of Slk19, Cse4, and cohesins (Richmond et al. 2013 Molecular Biology of the Cell 24:566-577; Mishra and Basrai 2019 Current Genetics 65:1325–1332; Mishra et al. 2019 Molecular Biology of the Cell 30:1020-1036; Mishra et al. 2016 Molecular Biology of the Cell 27:2286-2300). The authors have failed to discuss phosphorylation of Cse4 by Ipl1 and how it impacts kinetochore biorientation. This should be included in the context of Ipl1 (Boeckmann et al. 2013 Molecular Biology of the Cell 24:2034-2044).

Page 7 Lines 311-312. Authors stated “phosphorylated H2A recruits Sgo1 to the centromere”. This statement is context-dependent and is not completely correct considering the current state of literature in the field. Please elaborate this part from recent publications (Brimacombe et al. 2019 PLoS Biology 17: e3000331; Mishra et al. 2018 Cell Cycle 17: 11-23; Buehl and Kuo 2018 Current Genetics 64: 1215-1219).

Author Response

Title: The function and regulation of protein phosphatases in establishing chromosome bipolar attachment.

Authors: Delaney Sherwin and Yanchang Wang

In this manuscript, authors have reviewed the current state of literature on the role of protein phosphatases in regulation of chromosome bipolarity for the maintenance of chromosome stability. The objectives of this review are logical, timely, thought-provoking, and provide deeper insights into the current state of knowledge in the field. The manuscript is very well written and organized into sections that are easy to follow and understand for the broad readerships of the journal. Overall, manuscript is acceptable pending minor revisions outlined below.

Specific Comments:

Authors have very elegantly discussed the role of different kinases (for example Ipl1, Mps1, Cdk), but have omitted the role of polo-like kinase Cdc5, which also plays a critical role in kinetochore function. This review appears incomplete without information on Cdc5. Authors should add some information on this topic from recent papers on Cdc5-mediated phosphorylation of Slk19, Cse4, and cohesins (Richmond et al. 2013 Molecular Biology of the Cell 24:566-577; Mishra and Basrai 2019 Current Genetics 65:1325–1332; Mishra et al. 2019 Molecular Biology of the Cell 30:1020-1036; Mishra et al. 2016 Molecular Biology of the Cell 27:2286-2300).

Response: We highly appreciate the constructive comments from the reviewer. We totally agree that Cdc5 kinase is important for this review. As suggested, we added a new paragraph to discuss the functions of Cdc5 kinase in the regulation of sister chromatid cohesion. Line 150-163.

The authors have failed to discuss phosphorylation of Cse4 by Ipl1 and how it impacts kinetochore biorientation. This should be included in the context of Ipl1 (Boeckmann et al. 2013 Molecular Biology of the Cell 24:2034-2044).

Response: We added this information in the Ipl1 kinase section. Line 90-91.

Page 7 Lines 311-312. Authors stated “phosphorylated H2A recruits Sgo1 to the centromere”. This statement is context-dependent and is not completely correct considering the current state of literature in the field. Please elaborate this part from recent publications (Brimacombe et al. 2019 PLoS Biology 17: e3000331; Mishra et al. 2018 Cell Cycle 17: 11-23; Buehl and Kuo 2018 Current Genetics 64: 1215-1219).

Response: In the revised version, we discussed the role of Cse4-Sgo1 interaction in chromatin recruitment of Sgo1. Line 333-338.

Reviewer 2 Report

The manuscript by Sherwin and Wang describes the roles of and interplay between the many kinases and phosphatases that regulate cell cycle progression and the formation and dissolution of attachments between microtubules and kinetochores to ensure bipolar attachments and accurate chromosome segregation. The review is in general very clearly written and packed with information – mainly from budding yeast, but bringing in information from other systems as well. However, it could be greatly improved by an extreme makeover involving a change of title, reorganization of the text, and inclusion of figures to illustrate key points in the text. In addition, there are a few statements that strike me as going against current understanding. These either need to be backed up with citations or edited.

Major Comments:

The review would benefit tremendously by adding figures to illustrate key points made in the text. The text does a pretty good job of describing biorientation and chromosome segregation and the roles of a multitude of kinases and phosphatases in words alone, but it is rare to see a 14 page long review that doesn’t have a single figure.
The title of the review is “The function and regulation of protein phosphatases in establishing chromosome bipolar attachment,” yet most of the information in the review is about the roles of kinases in regulating KT-MT attachments and checkpoints. Typically, these statements are followed by a few sentences that could be summarized by, “there are phosphatases that antagonize these kinases.” It is impossible to talk about the phosphatases without first explaining the kinases they act in opposition to, but it is misleading to claim that the review is about mitotic phosphatases. It is about the antagonistic relationships between kinases and phosphatases and how their opposing functions promote accurate chromosome segregation. The title should reflect this.

One example: in lines 55-57, the introduction ends with the sentence, “In this review, we discuss the function and regulation of phosphatases in establishing chromosome bipolar attachment, with a focus on budding yeast Saccharomyces cerevisiae.” Line 58 is the heading for the next section: “2. Kinases Involved in the Establishment of Chromosome Bipolar Attachment,” which is followed by two full pages of discussion of kinases.

Another example: Section 3 is entitled, “Protein Phosphatases Involved in Chromosome Bipolar Attachment.” However, much of this section is devoted to more evidence of the importance of the kinases introduced in section 2. For example, lines 169-178 describe serine/threonine to alanine mutations that prevent phosphorylation and mimic mutations in the kinases. This supports the importance of the kinases. It is not evidence that a phosphatase plays an important role. Evidence for the importance of phosphatases only comes in lines 179-186. Thus, half of this long paragraph is really about the kinases. The information about S/T to A mutations would be better placed earlier, in the section about Ipl1. This problem is repeated in subsequent sections

Final example: on p. 5 of 14 in the Phosphatases section, Lines 197-201 are about kinases and 202-203 are about phosphatases but only say “it is unclear” what their role is. Lines 204-218 are about kinases, and line 219-222 are speculation about a possible role for phosphatases.

The speculation and discussion about what is and is not known is interesting and important, but it is odd to me to give the review the current title saying that it is about phosphatases.

Minor Comments:

Lines 27-28: The goal of mitosis isn’t to make sure that each daughter cell receives an “equal number of replicated chromosomes.” It is to make sure that daughters receive exactly one copy of every chromosome. Lines 34-35: most bipolar attachments form in prometaphase as chromosomes are aligning on the metaphase plate. I don’t think that spindles are elongating at this stage in most systems. If I am correct in my belief, spindle elongation does not “generate tension across sister kinetochores.” Either this statement should be revised or references should be included to support the statement. Lines 35-36: In my opinion, the SAC is active by default until bipolar attachments are made. Thus, it is not that “faulty kinetochore attachment activates the spindle assembly checkpoint” but that the SAC is active until every chromosome has made a bipolar attachment. This idea is well expressed in lines 46-47. Lines 96-97: “Mutation of these phosphorylation sites on Ndc80 (ndc80-7A) contributes to sensitivity to the microtubule depolymerizing agent benomyl” is a confusing sentence. I assume this means that ndc80-7a INCREASES sensitivity to benomyl. Lines 77-104: the effects of Ipl1-dependent phosphorylation of Dam/DASH complex proteins and Ndc80 seem confusing, with Dam1 phosphorylation inhibiting Ndc80 recruitment to kinetochores and destabilizing KT-MT interactions and Ndc80 phosphorylation seemingly stabilizing KT-MT interactions. It would be interesting to expand this section with more discussion of these opposing effects of Ipl1. Line 160: dephosphorylation of Dam1 was observed “at a narrow window.” Please state when in the cell cycle this window occurs. Lines 172-173 It seems strange to suddenly say “we” and “our” Lines 169-178: I think the S/T to A mutation section would be more effective in the kinase section than in the phosphatase section. A one sentence reminder could be included in the phosphatase section supporting the expected consequence of dephosphorylation. This comment applies not only here, but to all summaries of alanine mutations in the phosphatase section There is genetic data in a number of systems showing that Ipl1 and PP1 antagonize one another, for example that PP1 mutations can be suppressed by Ipl1 mutations or vice versa. It might be interesting to refer to some of this data to support the model that PP1 removes phosphate groups added by Ipl1. Line 251: “Fin1 protein localizes in the nucleus through metaphase” It might be helpful to remind readers that budding yeast undergoes a closed mitosis and the nuclear envelope does not break down. Line 304: typo PP2ARst1

Author Response

Response to the reviewer’s comments

The manuscript by Sherwin and Wang describes the roles of and interplay between the many kinases and phosphatases that regulate cell cycle progression and the formation and dissolution of attachments between microtubules and kinetochores to ensure bipolar attachments and accurate chromosome segregation. The review is in general very clearly written and packed with information – mainly from budding yeast, but bringing in information from other systems as well. However, it could be greatly improved by an extreme makeover involving a change of title, reorganization of the text, and inclusion of figures to illustrate key points in the text. In addition, there are a few statements that strike me as going against current understanding. These either need to be backed up with citations or edited.

Major Comments:

The review would benefit tremendously by adding figures to illustrate key points made in the text. The text does a pretty good job of describing biorientation and chromosome segregation and the roles of a multitude of kinases and phosphatases in words alone, but it is rare to see a 14 page long review that doesn’t have a single figure.

Response: We would like to thank the reviewer for the insightful comments for this review. We added four figures in the revised version.

The title of the review is “The function and regulation of protein phosphatases in establishing chromosome bipolar attachment,” yet most of the information in the review is about the roles of kinases in regulating KT-MT attachments and checkpoints. Typically, these statements are followed by a few sentences that could be summarized by, “there are phosphatases that antagonize these kinases.” It is impossible to talk about the phosphatases without first explaining the kinases they act in opposition to, but it is misleading to claim that the review is about mitotic phosphatases. It is about the antagonistic relationships between kinases and phosphatases and how their opposing functions promote accurate chromosome segregation. The title should reflect this.

Response: The title was changed to “The opposing functions of protein kinases and phosphatases at kinetochores promotes chromosome bipolar attachment”.

One example: in lines 55-57, the introduction ends with the sentence, “In this review, we discuss the function and regulation of phosphatases in establishing chromosome bipolar attachment, with a focus on budding yeast Saccharomyces cerevisiae.” Line 58 is the heading for the next section: “2. Kinases Involved in the Establishment of Chromosome Bipolar Attachment,” which is followed by two full pages of discussion of kinases.
● Response: Changed to: In this review, we discuss the antagonistic relationships between kinases and phosphatases and their roles in promoting accurate chromosome segregation, with a focus on budding yeast Saccharomyces cerevisiae.

Another example: Section 3 is entitled, “Protein Phosphatases Involved in Chromosome Bipolar Attachment.” However, much of this section is devoted to more evidence of the importance of the kinases introduced in section 2. For example, lines 169-178 describe serine/threonine to alanine mutations that prevent phosphorylation and mimic mutations in the kinases. This supports the importance of the kinases. It is not evidence that a phosphatase plays an important role. Evidence for the importance of phosphatases only comes in lines 179-186. Thus, half of this long paragraph is really about the kinases. The information about S/T to A mutations would be better placed earlier, in the section about Ipl1. This problem is repeated in subsequent sections

Final example: on p. 5 of 14 in the Phosphatases section, Lines 197-201 are about kinases and 202-203 are about phosphatases but only say “it is unclear” what their role is. Lines 204-218 are about kinases, and line 219-222 are speculation about a possible role for phosphatases.

Response: We have moved the description of the dam1-3A mutation to the Ipl1 kinase section. In addition, we added one sentence in the beginning of this paragraph: “The role of Dam1 dephosphorylation in KT-MT interaction has been studied using the phospho-deficient dam1-3A mutants, where three Ipl1 consensus sites are mutated to alanine.”

As suggested, we have moved part of the introduction about the function if Ipl1 in checkpoint control into the Ipl1 kinase section.

The speculation and discussion about what is and is not known is interesting and important, but it is odd to me to give the review the current title saying that it is about phosphatases.

Response: we hope the change of the title of this review will address this concern.

Minor Comments:

Lines 27-28: The goal of mitosis isn’t to make sure that each daughter cell receives an “equal number of replicated chromosomes.” It is to make sure that daughters receive exactly one copy of every chromosome.

Response: Changed.

Lines 34-35: most bipolar attachments form in prometaphase as chromosomes are aligning on the metaphase plate. I don’t think that spindles are elongating at this stage in most systems. If I am correct in my belief, spindle elongation does not “generate tension across sister kinetochores.” Either this statement should be revised or references should be included to support the statement.

Response: Changed to “The establishment of chromosome bipolar attachment or biorientation enables the onset of anaphase”.

Lines 35-36: In my opinion, the SAC is active by default until bipolar attachments are made. Thus, it is not that “faulty kinetochore attachment activates the spindle assembly checkpoint” but that the SAC is active until every chromosome has made a bipolar attachment. This idea is well expressed in lines 46-47.

Response: Changed to “The spindle assembly checkpoint (SAC) is active to delay anaphase entry until every chromosome has made a bipolar attachment.”

Lines 96-97: “Mutation of these phosphorylation sites on Ndc80 (ndc80-7A) contributes to sensitivity to the microtubule depolymerizing agent benomyl” is a confusing sentence. I assume this means that ndc80-7a INCREASES sensitivity to benomyl.

Response: As suggested, we have replaced “Contributes to” with “increases the”.

Lines 77-104: the effects of Ipl1-dependent phosphorylation of Dam/DASH complex proteins and Ndc80 seem confusing, with Dam1 phosphorylation inhibiting Ndc80 recruitment to kinetochores and destabilizing KT-MT interactions and Ndc80 phosphorylation seemingly stabilizing KT-MT interactions. It would be interesting to expand this section with more discussion of these opposing effects of Ipl1.

Response: Our description for the function of Ndc80 phosphorylation in KT-MT interaction is incorrect. The sentence was changed to: “In mammalian cells, Ndc80/Hec1 is also a substrate of Aurora B kinase and Ndc80 dephosphorylation promotes kinetochore-microtubule attachments.” We really appreciate that the reviewer has pointed out this mistake.

Line 160: dephosphorylation of Dam1 was observed “at a narrow window.” Please state when in the cell cycle this window occurs.

Response: We added “likely prior to anaphase onset”. We analyzed Dam1 phosphorylation in synchronized cells during cell cycle, but we did not analyze this phosphorylation in cells arrested at different cell cycle stages.

Lines 172-173 It seems strange to suddenly say “we” and “our”

Response: “We” and “Our” are deleted in the revised version.

Lines 169-178: I think the S/T to A mutation section would be more effective in the kinase section than in the phosphatase section. A one sentence reminder could be included in the phosphatase section supporting the expected consequence of dephosphorylation. This comment applies not only here, but to all summaries of alanine mutations in the phosphatase section

Response: As discussed above, we have moved the description of the identification of Dam1 phosphorylation sites to the kinase section. Because this is a special issue for “Protein Phosphatases and Cell Cycle Regulation in Yeasts”, we still keep part of this paragraph in the phosphatase section.

There is genetic data in a number of systems showing that Ipl1 and PP1 antagonize one another, for example that PP1 mutations can be suppressed by Ipl1 mutations or vice versa. It might be interesting to refer to some of this data to support the model that PP1 removes phosphate groups added by Ipl1.

Response: We added several references regarding the genetic interaction between ipl1 and glc7 mutants, which supports the model that PP1 reverses the phosphorylation imposed by Ipl1.

Line 251: “Fin1 protein localizes in the nucleus through metaphase” It might be helpful to remind readers that budding yeast undergoes a closed mitosis and the nuclear envelope does not break down.

Response: We added one sentence to emphasize that budding yeast undergoes a closed mitosis.

Line 304: typo PP2ARst1 

Response: Changed!